# Quality Assessment of a Novel Camera-Based Measurement System for Roughness Determination of Concrete Surfaces—Accuracy Evaluation and Validation

**DOI:** 10.3390/s22114211

**Published:** 2022-05-31

**Authors:** Barış Özcan, Jörg Blankenbach

**Affiliations:** Geodetic Institute and Chair for Computing in Civil Engineering & Geo Information Systems, RWTH Aachen University, Mies-van-der-Rohe-Str. 1, 52074 Aachen, Germany; blankenbach@gia.rwth-aachen.de

**Keywords:** concrete surface roughness, non-destructive testing, optical surface measurement, image-based 3D reconstruction, accuracy, comparative study

## Abstract

The roughness of a surface is a decisive parameter of a material. In rehabilitation of concrete structures, for example, it significantly affects the adhesion between the coating material and the base concrete. However, the standard measurement procedure in construction suffers from considerable disadvantages, which leads to the demand for more sophisticated methods. In a research project, we, therefore, developed a novel camera-based measurement system, which is customized to meet the prevailing requirements for practical use on construction sites. In this article, we provide an overview of the measurement system and present comprehensive examinations to evaluate the accuracy and to provide evidence of validity. First, we examined the accuracy of the system by empirically assessing both trueness and precision of measurements using three concrete specimens. Trueness was determined by comparing the surface measurements to those of a highly accurate microscope system, revealing RMSE values of around 40–50 µm. Precision, on the other hand, was assessed considering the scattering of the roughness measurements under repeat conditions, which led to standard deviations of less than 6 µm. Furthermore, to proof validity, a comparative study was conducted based on sixteen concrete specimens, which includes the sand patch method and laser triangulation as established roughness measurement methods in practice. The empirically determined correlation coefficients between all three methods were greater than 0.99, indicating extraordinarily high linear relationships. Among them, the greatest correlation was between the camera-based system and laser triangulation.

## 1. Introduction

The surface represents the interface of a material with its environment and, thus, is decisive in how the material interacts with its surrounding medium. The texture of a surface consists of different orders of shape deviations, such as form deviation, waviness and roughness, while the latter substantially influences the mechanical properties and, consequently, affects the performance of a material. Among others, roughness significantly contributes to friction, wear, adhesion or wettability of a material, to mention a few. 

In building construction, a main application field can be found in the repair, protection and rehabilitation of degraded building components, such as concrete structures. After an initial investigation of the defects in order to determine the cause of deterioration, the substrate is prepared to remove loose or cracked concrete. Following this, corrosion protection for visible reinforcement and concrete replacement system (coating material) is applied to the flaws and chipped areas in the concrete surface. In this process, the applied liquid coating material flows into the gaps and holes of the base concrete, hardens and anchors like dowels or snap fasteners. However, to ensure the bond between coating material and base concrete, the concrete surface requires certain adhesive strength [1,2,3,4,5]. Recent experiments by Fan et al. confirmed that an increased roughness and lower age of old concrete can improve the bond of new-to-old concrete structures, where the effect of roughness was more significant than that of the age of the old concrete [6]. 

Another application can be found in pavement construction. In this context, roughness significantly contributes to the mechanical properties of road surfaces, which has a great impact on traffic conditions. For example, roughness substantially influences frictional resistance between vehicle tires and pavement, which in turn affects the propulsive and brake force of vehicles, hence significantly contributing to traffic safety. In addition, fuel consumption, rolling resistance, tire wear and tire and road noise, just to mention some, also depend considerably on road roughness. 

The most frequently used and standardized method so far for estimating roughness of concrete surfaces is the sand patch method according to Kaufmann [7]. This procedure involves a pre-defined amount of sand to be applied on the surface to be measured and distributed evenly in circular movements. Finally, the diameter of the resulting circle permits drawing conclusions of the roughness of the surface. Due to its simplicity in the application and low equipment requirements, the method gained huge acceptance in practice over years. However, it involves several disadvantages. First of all, it is not applicable on highly slanted surfaces or ceilings. In addition, the outcome of the method is considerably affected by user experience, and typically, it does not provide reproducible results. Although there are more sophisticated measurement systems, for example based on laser triangulation or microscopy, they are associated with shortcomings as well. They are often too expensive, require a complex or non-portable measurement setup or are in general not feasible for deployment on construction sites. 

In a research project, we, therefore, developed a novel camera-based measurement system for roughness determination of concrete surfaces. The measurement system is customized to meet the prevailing requirements for practical use on construction sites and, thus, comes with several advantages compared to other systems. First of all, it is digital and enables a fully automatic analysis of the surface. In addition, due to its usability, no in-depth training of the operator is necessary. The camera-based approach further facilitates a large-scale and area-based assessment of the surface, which is mandatory for capturing surface features properly. The optical and thus non-contact measurement method guarantees non-destructive testing. The photogrammetric procedure used for 3D reconstruction is further known to provide high-resolution and highly accurate measurements. The requirements to the hardware components can mainly be reduced to an industrial camera and a cross slide, keeping the system relatively low-cost and lightweight. Furthermore, the system is portable and equipped with rechargeable batteries, which enables flexible deployment on construction sites, including arbitrary oriented surfaces, such as walls or ceilings. To our knowledge, there is currently no other system combining the features outlined above, a fact that emphasizes the novelty of our proposed measurement system. A more detailed description of the system is given in [8,9,10], including an overview of the hardware setup, the software pipeline for 3D reconstruction, experiments regarding camera calibration and first results for roughness measurements. 

In this article, we build on our previous work and provide further examinations for the evaluation of measurement quality of the proposed measurement system. For this purpose, two kinds of investigations are carried out. The first kind of investigation deals with the accuracy evaluation of the measurement system. This involves evaluating both trueness and precision of measurements. Trueness is determined by comparing the measurements to those of a high-accurate 3D scanning microscope. Precision, on the other hand, is determined by assessing the repeatability of measurements under repeat conditions. The second type of investigation covers the validation of the system. This is conducted by a comparative study with established roughness measurement methods, such as the sand patch method and laser triangulation. In the experiments, three small respectively sixteen large concrete slabs were used as specimens. 

Following this introduction, the paper is organized as follows: Section 2 reviews recent advances in methods for roughness determination of concrete surfaces and additionally provides a terminology of quality assessment of measurement systems. In Section 3, we briefly summarize the camera-based measurement system, including the hardware and software components. Section 4 presents the examinations carried out on the measurement accuracy. This involves experiments for assessing both trueness and precision. Subsequently, in Section 5, a comparative study to established roughness measurement methods is provided. Section 6 discusses the results and, finally, in Section 7, we present our conclusions with an additional insight into future work. 

## 2. State of the Art and Related Work

### 2.1. Recent Advances in Roughness Measurement Methods

Throughout the years, researchers proposed several systems and methods for determining the roughness of construction materials. However, none of them have proven to be the predominant method in this field, as each of them is designed to meet specific requirements and, in general, all methods are also associated with shortcomings. 

In this regard, Santos et al. provide a comprehensive review of the state-of-the-art on roughness quantification methods for concrete surfaces [1]. Among these, the authors divide the methods into contact and non-contact methods, which in turn are subdivided into qualitative (assessment by humans) and quantitative (assessment by means of a number). In the following, we survey a selection of the most recent developments and most relevant research regarding surface roughness assessment and highlight their features. 

In qualitative assessment, Concrete Surface Profile (CSP), which is a standardized measure by the International Concrete Repair Institute (ICRI), often comes into use [11]. This method involves a qualified operator to perform a visual inspection by comparing the target surface with reference samples. The reference samples are a set of ten concrete slabs with different degrees of roughness. The major drawback of this method, however, is the influence of the operator’s opinion, since a subjective assessment is conducted. 

As for contact methods, stylus profilometers belong to the most popular instruments for measuring surface roughness. This method involves a very thin stylus tip consisting of sapphire or diamond to be dragged across the surface. The movement of the stylus tip is tracked on a computer, which derives 3D coordinates of the surface. The tip usually has a canonical shape of a sphere typically with a radius of 2 µm, 5 µm or 10 µm. Accordingly, grooves narrower than the radius of the stylus tip cannot be measured. Kubátová et al. found that the impact of size and shape of the tip is under 3% for primary profiles and under 1% for roughness profiles [12]. Nevertheless, the difference between new and old styli was more severe, being as much as 20% for primary profiles and about 5% for roughness profiles. Additionally, Grochalski et al. examined the scattering of roughness parameters based on four different stylus profilometers, two of them with a skid and the other two skidless [13]. The experiments revealed that there is a considerable difference between styli with skid and skidless ones, while the skidless ones had a better resolution leading to lower scattering of measured values. Accordingly, the measurements depend highly on physical properties of the stylus tip, which toughens the comparability of different systems. Another well-known issue of stylus profilometers is the limitation to line-based measurements. After all, a single profile line is not able to cover all relevant surface features. For example, measurements on surfaces with periodic creases should not be performed parallel to the direction of the creases; otherwise, the actual maximum height of the surface cannot be detected. The method further is inappropriate for measuring adhesive surfaces and soft samples. Finally, contact-based measurements can lead to wear of both the measurement device and the measured surface, and thus, change the original structure of the surface. This, in turn, can lead to falsified results. 

For these reasons, non-contact methods hold a considerable advantage in comparison. Particularly, for characterization of surface texture, optical 3D profilometers are getting more and more recognition. These can essentially be divided into active and passive sensors. While active optical sensors are equipped with a light emitting source, which projects light onto the surface to be measured, passive optical sensors utilize naturally surrounding light. In both cases, however, the light is reflected by the surface and is captured by a light sensitive sensor. Fringe projection, laser triangulation or 3D scanning microscopy based on interferometry, focus variation or confocal laser scanning are some examples for recent optical methods deployed for surface texture characterization. Although these methods fundamentally capture the surface in an area-based manner (in the case of laser triangulation, a minor adjustment is required to achieve this), surface texture is nowadays still mostly characterized based on 2D profile lines. 

Čairović et al. compared a laser triangulation system and the sand patch method for assessing surface roughness parameters [14]. For their experiments, the authors used five concrete specimens with roughness ranging from very smooth to rough and illustrated the correlation of both methods in a diagram. Furthermore, on the basis of their investigations, they confirm that the sand patch method does not provide any detailed information regarding topology. Additionally, for very smooth surfaces, they propose to use less volume of sand (i.e., 5 mL instead of 25 mL). However, the sand patch method was sufficient for categorization of the surface according to the fib Model Code 2010 [15]. 

A microscope system was utilized by Schabowicz et al. in order to assess concrete roughness in expansion gaps [16]. For this purpose, the InfiniteFocus G5 microscope system, which is based on focus variation, was deployed. As objects of investigation, they used three cubic concrete specimens with a side length of 100 mm and treated one particular side of each specimen successively with two different modifications: manual cleaning using a wire brush and mechanical cleaning using a rotary cleaning tool followed by a cutting tool. Before treatment as well as after every modification, an area of 99 × 99 mm of the surface was captured by the system with a resolution of 3162 × 3162 measurement points. Based on these measurements, several surface features, such as histograms, Abbott-Firestone material ratio curves and microasperity profiles, were derived. In addition, area-based parameters for surface topography, as for instance height, frequency and hybrid parameters, were also determined. However, parameters characterizing shape and dimensions of microasperities were derived based on extracted surface profiles rather than from the area. They conclude that the surfaces after treatment with mechanical cleaning led to increased mean roughness of around 4–5 times compared to the original surface. In contrast, manual cleaning led in two cases only to slight increasements of mean roughness and in one case even to lower mean roughness compared to the original surface. The reason for lower roughness is most likely due to smoothing out peaks and valleys of the surface by hand cleaning.

Usually, measurements of surface topography performed by microscope systems are relatively time-consuming due to the large amount of data that is captured and processed. To tackle this issue, Gladines et al. [17] proposed a two-step pipeline for shape-from-focus (SFF) 3D profilometry. Their key idea is to perform initially a coarse measurement by a faster profilometry technique to obtain a rough estimate of the object surface. This, in turn, is used to define a margin around the estimated depth range for the finer measurement using the SFF technique. The results showed that this approach can lead to a time saving of more than 40%. 

Although optical 3D profilometers, as non-contact methods, hold considerable advantages compared to contact methods, they are associated with some limitations as well. Crucial challenges are typically reflective, glossy and translucent surfaces. Furthermore, they often come in a complex or non-portable setup and are commonly quite expensive. A comprehensive review on the challenges of the state-of-the-art active 3D optical measurement techniques is provided in [18]. One of their key statements is that modern 3D optical metrology methods are based on the one-size-fits-all approach or often require prohibitively expensive customizations to meet the requirements for a specific task.

### 2.2. Terminology of Measurement Accuracy and Literature Review on Accuracy Assessment

The true value of a quantity can never be determined exactly by measurements, since measurements are always affected by uncertainties. This matter of fact results in measurement deviations or measurement errors respectively, which must, therefore, be quantified to assess the accuracy of a measurement method.

In metrology, traditionally, a distinction is made between gross errors as well as systematic and random measurement deviations (also called systematic and random errors). Gross errors are errors in the true sense of the word and show typically large error values such as outliers. They are caused, for example, by human errors of the observer, an unsuitable measurement method or a defective measuring instrument. Systematic measurement deviations, on the other hand, influence the measurement result in a regular manner and are caused by the imperfection of the measuring instrument or by the non-consideration of physical laws. What remains are the random measurement deviations, which are unavoidable and unpredictable according to their nature. In the Guide to the Expression of Uncertainty in Measurement (GUM) [19], this view is broadened by describing systematic errors of measurements mathematically in the same way as random errors instead of splitting the uncertainties into a known and unknown proportion. 

For the accuracy assessment of a measurement system, these different types of deviations have to be taken into account. Whereas gross errors and systematic deviations can be controlled, for example by high care and calibration, random errors always occur and can only be described by statistical methods. However, there is no standardized way to identify these systematic and random errors. Depending on the field, there are even different standards and guidelines for the specific definition of accuracy and for procedures of its assessment. In mechanical engineering, for example, the above-mentioned GUM is often used, whereas in other domains, the classical distinction between gross, systematic and random errors is predominant. 

In this paper, we refer to the definition according to ISO standard 5725-1 [20], which is relevant in metrology, to assess the accuracy of our measurement system. Measurement accuracy in general refers to the extent to which a measurement result approaches the desired true value of a measured variable. According to ISO 5725-1, the accuracy consists of the two aspects precision and trueness. Trueness describes the extent to which the expected value is approximated to the true value of the measured variable, while precision describes the extent to which measured values are scattered under repeat conditions. Random measurement deviations, therefore, primarily influence precision, while systematic effects act on trueness. The accuracy, thus, includes both the systematic and random deviations [21] and only if both are getting small, the measured values get close to the physically true value, and thus, the accuracy becomes high (see Figure 1).

For accuracy evaluation in practice, the expected value can be approximated by the arithmetic mean of measured values under repeat conditions. The unknown true value can be approximated as a normal comparative standard by a reference value determined with a highly accurate measuring instrument. The precision can then be quantified by analyzing the deviations between the mean value and the single measurements (e.g., by an empirical standard deviation), while the comparison of the deviations between the reference value and the average value from a series of measurements, often referred to as absolute accuracy, can be used for assessment of the trueness.

The literature on the evaluation of measurement accuracy respectively quantification or reduction of measurement errors shows a variety of approaches. Haitjema [22], for example, dealt with issues occurring when topographic parameters (such as roughness parameters) are derived from surface measurements, which are known to be subject to uncertainties. To solve this, a method for estimating uncertainties in the derived parameters is presented, and furthermore, the influencing factors in the uncertainty estimation, such as amplification, spatial resolution or probe tip radius, are exemplary assessed based on three different roughness parameters. 

Two different methods for noise reduction in coherence scanning interferometry (CSI) were evaluated in [23]. One of them is averaging a sequence of repeated topography measurements and the other one is increasing the sampling frequency of the fringe signal during a single data acquisition (also known as oversampling). One of their key findings is that the measurement noise is rising with increasing surface tilt and height variation. Furthermore, the effects of both noise reduction methods were compared based on surface topography measurements in the presence of environment-induced vibration. The results showed that the averaging method is effective for reducing all sources of noise, including environment-induced vibration and camera noise. The signal oversampling method, on the other hand, had the same noise reduction effect, but in the presence of vibration, this was valid only when the surface was flat with zero tilt. 

Interesting approaches for detection and suppression of high-frequency measurement errors in surface topography measurements were presented in [24]. For this purpose, different techniques for detection and for suppression (such as Power Spectral Density (PSD) or autocorrelation function analysis, spline modifications of profiles and shape analysis of the autocorrelation function) were compared based on several kinds of surfaces textures. The author further discusses the difficulties encountered by each method for the different surface textures.

The precision of shape-from-focus imaging was examined regarding texture frequencies and window sizes of a focus measure in [25]. According to their experimental results, a smaller window size is not sufficient for a correct focus measure. In contrast, a window size that is approximately equal to a pixel-cycle length of the texture was able to provide better precision. 

## 3. Measurement System

### 3.1. Apparatus

The hardware of the measurement system consists mainly of a mechanical cross slide with controlling unit and propulsion, which guides a monocular industrial camera over the surface to be measured. The camera is moved in a meandering trajectory with a viewing direction perpendicular to the surface. Strongly overlapping images with around 60–80% overlap are captured and continuously transferred to a laptop by USB. The deployed camera is equipped with a monochromatic sensor featuring a resolution of 10 megapixels. However, since the camera sensor is based on rolling shutter technology, the images are captured in stop-and-go mode. The total time for capturing a surface entirely depends primarily on the total number of images to be taken, and thus, on the configured degree of overlap in the images, and currently takes approx. 5 min for 30 images. 

Four LED strips are attached all around the system for a diffuse illumination of the surface to be captured, which enables measurements independent of ambient light. Furthermore, to ensure a mobile and flexible deployment, the system is equipped with rechargeable batteries. To keep the measurement system dust- and waterproof, the entire hardware assembly is embedded in a black casing. Additionally, two ergonomic handles are attached to the sides of the casing to facilitate measurements on arbitrary oriented surfaces. Figure 2 presents images of the camera-based system. 

### 3.2. Software

In order to assess the surface roughness, a fully automated software workflow was developed that operates on the images captured by the measurement system. The software first reconstructs a 3D point cloud of the surface based on a two-step image matching pipeline: Structure from Motion (SfM) is used to recover the camera poses, and building on that, Dense Image Matching (DIM) is performed to reconstruct a dense point cloud of the object surface. For SfM, we make use of the open source library OpenCV [26], while we implemented DIM onto graphics processing units (GPUs) without third-party libraries. Finally, the generated dense point clouds are utilized to derive roughness parameters of the surface. An overview of the single steps for the roughness determination procedure is provided in Figure 3. Additionally, the following subsections go into further detail for the fundamental steps of the workflow.

#### 3.2.1. Structure from Motion

Initially, the SURF feature detector [27] is utilized to identify distinctive points, also called feature points, in the images. Subsequently, matching of these points in distinct images is performed to both identify overlapping image pairs and to compute the relative orientation of these pairs, represented by a rotation matrix and a translation vector. 

Starting from an initial image pair and the respective relative orientation, matched feature points are triangulated in space, which results in the associated 3D object points in a local coordinate system. Subsequent images are incrementally registered in the present coordinate system utilizing 2D-to-3D point correspondences of 2D feature points in the images to be registered and corresponding 3D object points triangulated previously. After each image registration step, incremental bundle adjustment is applied in order to optimize the 3D object point coordinates, the camera pose for each image consisting of 6 degrees of freedom for position and orientation and the feature point coordinates simultaneously. Figure 4 provides a simplified overview of the SfM procedure.

#### 3.2.2. Dense Image Matching

Based on the recovered camera poses, DIM is performed for matching every individual image point in the image pairs in order to obtain a dense point cloud of the object surface. We implemented semi-global matching (SGM) [28] for generating disparity maps of image pairs, which basically encode the offset of corresponding image points in these pairs. These disparity maps are further utilized to derive the actual depth of the object points, which are associated to the image points. 

Since DIM is a quite compute-intensive algorithm and requires considerable computing time on central processing units (CPUs), we parallelized the sub-algorithms of SGM and implemented for graphics processing units (GPUs) for parallel computing. As programming platform, compute unified device architecture (CUDA) [29] is used, which enables to implement algorithms specifically for GPUs of the manufacturer NVIDIA.

#### 3.2.3. Roughness Assessment

A frequently used roughness parameter in practice is the arithmetical mean height parameter *Ra* [30]. Traditionally, this parameter is assessed in a line-based matter (in particular because surfaces are still often captured with line-based methods, such as stylus profilometers) and basically describes the mean absolute deviation of the captured profile line to a mean line. In case of area-based assessment, however, the parameter is also typically referred to as *Sa* [31]. Since we have three-dimensional point clouds at hand instead of two-dimensional lines, we consequently adapt this parameter for our particular case. This is done by first estimating a mean plane $\hat{\pi }$ through all points in the point cloud using normal equations: (1)$$\hat{\pi }=\left[\begin{array}{c} \hat{a}\\ \hat{b}\\ \hat{c} \end{array}\right]={\left(A^{T}A\right)}^{-1}A^{T}b,$$
with design matrix A containing mainly the position coordinates x, y of the points, response vector b containing the height coordinates z of the points and the estimated parameters $\hat{a},\;\hat{b},\hat{c}$ for an equation of a plane in the form of ax+by+c=z. 

Finally, the roughness parameter *Sa* is determined by calculating the mean vertical distance of all points in the point cloud to the estimated mean plane: (2)$$Sa_{pointcloud}=\frac{1}{n}\sum_{i=1}^{n}d_{vertical}\left(p_{i},\hat{\pi }\right)=\frac{1}{n}\sum_{i=1}^{n}{\hat{e}}_{i},$$
with points $p_{i}$ of the point cloud, the function $d_{vertical}$ for the vertical distance of a point to the estimated mean plane $\hat{\pi }$ (also referred to as residual ${\hat{e}}_{i}$) and the total number of points n. 

## 4. Accuracy Evaluation

### 4.1. Study Design

To obtain information about the quality of the determined roughness results, respective analyses concerning the measurement accuracy are necessary. In this paper, we adhere to the standard ISO 5725-1 [20], which defines the accuracy of a measurement as a composition of two classes, systematic errors and random errors (see Section 2.2). Consequently, in this study, we divide the evaluation of the measurement accuracy into two groups. The first kind of investigation is dedicated to the assessment of the absolute accuracy, which in particular allows a conclusion about the trueness. This is performed by comparing the reconstructed point clouds of our measurement system directly with surface measurements of a reference system. The second kind of investigation deals with the assessment of the precision considering repeatability. This, in turn, is done by performing multiple measurements of the same object surfaces under repeat conditions and considering the scattering of the determined roughness values. For this study, three small-sized concrete specimens are used. 

### 4.2. Test Specimens

A set of three concrete specimens was custom-manufactured to serve as investigation objects. The slabs were designed and fabricated with a size of 15 cm × 15 cm in length and width and have a thickness of approx. 5 cm. All three slabs are pieces of the same concrete cast manufactured with a compressive strength class of C20/25 and a round quartz aggregate. However, in order to obtain specimens with different degrees of surface roughness, all three of them were treated with different types of surface retarders. Surface retarders are liquid substances that can be applied both in negative or positive process to create architectural exposed aggregate concrete. This method is known to produce reasonably accurate washout depths. In our case, the AMITOL S3/10, S3/80 and S3/300 were applied in a negative process, which lead to surfaces with washout depths of approx. 1.5 mm, 3.0 mm and 6.0 mm. In Figure 5, images of the specimens are shown, and in Table 1, their specifications are provided.

### 4.3. Measurements with a Reference System

To sufficiently estimate the accuracy of a particular measurement system, a reference system is required, which provides in general a superior accuracy respective to the compared one, preferably multiple times more accurate or even better. In particular, for our conducted accuracy examinations, a measurement system was required as reference that senses the surface in a non-destructive manner, with particularly high resolution and as complete as possible. Optical systems are known to commonly meet these requirements. Hence, as reference system, we deployed an optical 3D measurement microscope based on focus variation technology. 

In surface metrology, the term Focus Variation (FV) refers to optical systems (usually a microscope) that exploit a limited depth of field (DOF) to obtain 3D information of an object surface. The sensor head of the measurement system is driven in the direction of the optical axis (typically towards the object surface) and a stack of images is captured continuously. Due to the limited DOF, only small regions of the object surface are captured sharply. Considering a sharpness measure for each image point, the depth of the associated object point can be derived. In this way, high-resolution surface measurements are achieved in vertical direction and, by movement of the sensor head across the surface, also in lateral direction. Furthermore, due to the photosensitive sensor, true color information is obtained for each object point as well. However, the method has some requirements for the object surface, such as structure and opacity. 

Particularly, for reference measurements on our three concrete specimens, we used the Alicona InfiniteFocus G5 microscope with a 5× objective. Figure 6 presents an overview of the measurement configuration during the capture of a particular specimen. In addition, Table 2 provides the most relevant specifications of the system with the deployed objective.

From each concrete specimen, a section with a lateral size of around 52.5 mm × 52.5 mm was captured using the microscope system. This led to measurement times of around 1–2 h for each specimen. The resulting surface meshes are in STL-format and consist of around 80 million faces with around 40 million vertices. However, since the surface can only be captured (almost) exactly vertically due to the measurement principle, some gaps occur in the digitized surface, especially in the presence of stronger overhangs. The obtained meshes of all three specimens are shown in Figure 7 from top and oblique view. 

### 4.4. Measurements with Our Camera-Based System

In order to be able to evaluate both the absolute accuracy and the reproducibility, we captured each of the three concrete specimens 10 times with our camera-based system under repeat conditions. To perform independent measurements and to simulate more practical conditions, after each individual measurement, the system was lifted up and repositioned roughly at the same location on the specimen. The surface was captured with an image overlap of around 78% in x-direction and 69% in y-direction, resulting in a total of 30 images. This particular configuration leads to around 5 min for one capture run. The image data are subsequently processed by the software for 3D reconstruction (see Section 3.2). The outcoming dense 3D point cloud of the object surface typically consists of around 15 million points and also contains greyscale values of the points. Figure 8 shows exemplarily one point cloud generated for each of the concrete specimens in top and oblique view. In addition, for clarity, rectangles are drawn in the figure to indicate the sections in the point clouds that were also captured using the microscope system. 

### 4.5. Evaluation of Absolute Accuracy

To estimate the absolute measurement accuracy (=trueness) of our system, the measurements of both systems were comparatively evaluated. Specifically, the meshes obtained by the microscope system serve as reference models to which the deviations of the points in the point clouds were determined. For this examination, a single point cloud for each specimen was selected from the total set of generated point clouds and comparisons were performed to the reference meshes.

However, since the point clouds and the meshes were generated using different measurement systems, they are initially located in distinct coordinate systems. Consequently, a co-registration has to be performed in order to align both entities. This is done by first performing a rough alignment by picking pairs of points from the point cloud and corresponding vertices from the mesh. Using these pairs, a rigid body transformation is applied to roughly align the point cloud to the mesh. After the rough alignment, Iterative Closest Point (ICP) is applied for a best fit registration of the point cloud to the mesh. This is carried out using 50,000 randomly selected sample points from the point cloud. However, since there are gaps in the meshes, which could distort the registration, we used the option to iteratively remove the farthest points during the ICP process. After the fine alignment, the point clouds are manually cropped to the area that overlaps with the meshes, since they cover a larger area (100 mm × 100 mm) of the surface compared to the meshes (52.5 mm × 52.5 mm). This is necessary, since otherwise, some of the points in the point cloud would lead to an excessively high and incorrect distance due to missing corresponding triangles in the mesh. Finally, the point-to-mesh deviations can be determined. 

As quality measure, we make use of the root mean square error (RMSE). This is calculated by squaring the distances of the single points to the nearest triangles in the mesh, subsequently determining the arithmetical mean and finally taking the square root. Formally, this is defined as: (3)$$\mathrm{RMSE}=\sqrt{\frac{\sum_{i=1}^{n}dist{\left(p_{i},t_{i}\right)}^{2}}{n}},$$
with points $p_{i}$ from the point cloud, corresponding nearest triangles $t_{i}$ from the mesh, the function dist for the closest distance from a point to a triangle and the total number of points n. For the three point clouds of the concrete specimens, the RMSE results are provided in Table 3. 

Furthermore, for a visual evaluation, the point deviations of the point clouds to the meshes are depicted using a false color map. Figure 9 shows the deviations of the three point clouds to the meshes in false colors from top view. To the right of the color maps, there are also histograms revealing the distributions of the point deviations. The color red represents a positive deviation, blue a negative deviation and yellow/green only a minor or no deviation.

In all three cases, the distributions approximate roughly a normal distribution. In addition, no significant skew of the distributions can be observed, and the arithmetical mean values are located close to 0, which indicates unbiased measurements of the surfaces. Thus, systematic errors are likely to be absent. 

### 4.6. Evaluation of Precision

To evaluate the repeatability (=precision) of the system, the 10-fold repeated surface measurements (specifically the reconstructed point clouds) of all three concrete specimens and the roughness values derived from these point clouds were used. 

The precision of the roughness determination was assessed by calculating the arithmetical means and standard deviations of the ten derived roughness values for each particular specimen. Figure 10 shows the results in a bar chart. Each bar represents the arithmetical mean value of the ten measurements for each particular specimen. Furthermore, the standard deviations of the measurement series are indicated by error bars. For specimen 1, the mean value for roughness parameter *Sa* results in 0.1961 mm, for specimen 2, it results in 0.7879 mm and for specimen 3 it is 1.0782 mm. The standard deviations lead to 0.0014 mm for specimen 1, 0.0058 mm for specimen 2 and 0.0051 mm for specimen 3. 

Furthermore, to be able to expose suspicious irregularities in the repeated surface measurements, comparisons were performed between the reconstructed point clouds from the same specimens. For this purpose, pairs of point clouds for each specimen were formed and the point deviations between these pairs were determined. 

The default way to compute distances between two point clouds is the nearest-neighbor distance. In this method, for each point in the compared point cloud, the closest point in the reference point cloud is searched and the Euclidean distance is calculated. However, although both the compared and the reference cloud are samples from the same surface, in general, they consist of different sampling points. This, in turn, leads to rather imprecise and larger distances when simply considering point-to-point distances. Therefore, we use the option to additionally take neighbors of the closest point into account to locally fit a quadratic model to these. The distance is then calculated to this model instead of to the single closest point. 

In Figure 11, false color maps show the point distances exemplarily for one pair of point clouds for each specimen. To the right of the color maps, there are also histograms of the point distances, which indicate folded normal distributions. In each color map, the single point distances look rather similar, showing no particular irregularities, apart from slightly larger point distances at the borders of the exposed aggregates (particularly in the two rougher specimens). 

## 5. Validation

### 5.1. Study Design

In order to facilitate benchmarking for roughness determination and to provide confidence especially for practitioners, comparisons are required with established roughness measurement systems. Accordingly, a comparative study has been conducted with the sand patch method and a laser triangulation system. The measurement results for roughness obtained by these two methods serve as reference values in order to compare with the roughness results of the camera-based system. For this study, 16 large-sized concrete specimens serve as investigation objects, which were assessed with all three methods. 

### 5.2. Test Specimens

The 16 concrete specimens used in this study were manufactured with a lateral size of approx. 1.63 m × 0.70 m. Different roughening methods and concrete mixtures were combined in order to create individual surface characteristics for all 16 slabs. For instance, half of the slabs were fabricated with round-grained quartz aggregate and the other half with broken basalt aggregate. In addition, the slabs were manufactured with two different compressive strength classes, C20/25 and C50/60. As roughening procedure, four different methods came into use. Four slabs each were treated with either sand-blasting (blast furnace slag to be precise), water jetting, concrete milling or stock chiseling. The water-jetted surfaces were roughened with a pressure of 1700 bar; the sand-blasted surfaces, on the other hand, were roughened with an air pressure of 4 bar and an abrasive between 0.5–1.4 mm in diameter. Table 4 summarizes the individual concrete mixtures and roughening procedures for each slab.

### 5.3. Measurement Procedures

For assessment with the sand patch method, each concrete specimen was divided into six sections on which the measurements were performed. Precisely, on each section, the method was carried out four times, with a rotation of around 45 degrees after each execution (up to 135°), leading to a total of 24 individual measurement values for each specimen. Furthermore, to minimize user influences, the same qualified operator carried out all measurements. The arithmetical mean of the 24 individual values finally represents the reference value for the roughness of the respective specimen. The amount of sand used was 24 g. 

Measurements with the laser triangulation system were carried out in a similar manner, performing multiple measurements on the same sections as done with the sand patch method. The deployed ELAtextur [32] scans the surface with a laser on a circular path and based on this profile roughness is determined. However, unlike the previous procedure, only three measurements were performed on each section, with the instrument rotated approx. 90 degrees after each run. Although the device provides two different parameters for roughness, mean profile depth (MPD) and estimated texture depth (ETD) (ISO 13473-1- [33]), they are basically related by a simple linear function. Therefore, only one of them, specifically ETD, which is largely comparable to the mean texture depth (MTD) from the sand patch method (according to [34]), will be considered in the following examinations. In this case, again, the arithmetical mean of the measurement series (consisting of 18 values) serve as reference value for the roughness of the particular specimen. 

In contrast to both previously described measurement procedures, the measurements with the camera-based system were not performed exactly on the same predefined sections, but on three relatively central positions on the left, central and right side of the concrete specimens. Similar to the evaluation of accuracy in Section 4, each concrete surface was captured with high image overlap of around 70–80%, leading to a total of 30 images and resulting in point clouds with around 15 million points for each measurement run. The roughness value of a particular specimen is once again defined by the arithmetic mean of the measurement series, in this case consisting of three values per series. Figure 12 illustrates exemplarily the measurement setups with all three deployed methods on test specimens. 

### 5.4. Evaluation of Validity

Results for the 16 specimens obtained with all three methods are summarized in Table 5 and additionally illustrated in a bar chart in Figure 13. Generally, the roughness results obtained by the sand patch method scatter around the roughness results obtained by the laser triangulation system. However, for very rough surfaces, as for example in case of specimens 5 to 8 (water-jetted specimens), the sand patch method rather tends to overestimate roughness compared to the laser triangulation system. In contrast, the camera-based system shows an underestimation of roughness for each individual specimen compared to both other methods. However, these discrepancies, especially in case of the camera-based system, occur because the methods primarily provide different parameters for roughness. Nevertheless, a strong degree of relationship can be observed between the obtained roughness results respectively methods. 

Since all three methods provide different roughness parameters in principle, a direct comparison of the measurement results is hindered. Therefore, we determined the correlation coefficients between the obtained roughness results in order to estimate the linear relationship between all three methods. In Figure 14, the correlations between the methods are indicated in scatterplots. 

The correlation coefficient between the sand patch method and the camera-based system leads to 0.9928, indicating a very strong linear relationship. The linear relationship between laser triangulation and the camera-based system is even slightly stronger with a correlation coefficient of 0.9934. In contrast, the correlation between the sand patch method and laser triangulation is the weakest of the three, although it is still very strong with a correlation coefficient of 0.9917.

## 6. Discussion

The results of the accuracy evaluation and of the comparative study show that the proposed camera-based system provides measurements of sufficient high quality. Particularly, the high accuracy of the system (in both parts, trueness and precision) confirms reliable, accurate and reproducible measurements. In addition, the empirically determined correlation coefficients of the roughness measurements between the camera-based system to both established roughness measurement methods, sand patch method and laser triangulation, indicate almost perfect linear relationships, which further strengthens the confidence in the system. 

In our previous work [8,9,10], we provided a detailed introduction to the system, presented experiments for camera calibration and provided the first results for roughness determination. The findings in this paper complement and extend our previous work by presenting comprehensive examinations for evaluating the accuracy and confirming validity. The results show that the system provides highly accurate and reproducible measurements, which is necessary for deployment on construction sites for measuring surface roughness of concrete structures. 

However, the proposed measurement system also holds some limitations. The deployed industrial camera, for example, possesses a macro objective, which is known to have a small DOF and can, therefore, capture only a limited range of depth in focus. However, empirical investigations were conducted, revealing the actual DOF of the camera to be around 6 cm, which is typically more than sufficient for most concrete surfaces. The photogrammetric approach furthermore requires some conditions for measured surfaces. For example, transparent or reflective surfaces can be challenging, which, however, can be bypassed by preparation of the surface, as, for instance, by applying a thin matt coating to the surface. Additionally, both algorithms SfM and DIM require an irregular and non-repetitive pattern on the surface in order to guarantee unique matching of image points. DIM, especially the implemented SGM algorithm, is also in general compute-intensive, requiring a measurement computer with sufficient processing power. However, to decrease the total processing time, some of the compute-intensive sub-algorithms were parallelized and outsourced to GPUs. In addition, although the results in this paper confirm the measurement system to have an accuracy sufficient high for the field of civil engineering, in some other fields, higher accuracies and resolutions are required. In mechanical engineering, for example, technical components often have to be measured with an accuracy and resolution in the range of nanometers. In such cases, the use of other measurement systems, for example those based on microscopy, is recommended. 

For the conducted accuracy examinations, a 3D scanning microscope was deployed as reference system, which provides very high-resolution and highly accurate surface measurements. However, it also involves some significant limitations. For example, as it is generally known for optical systems, reflective or translucent surface structures, such as mineral aggregates like gravel, sand or natural stones, cannot be captured at all or only very inaccurately. In addition, the surface scanning of the microscope system is limited to the vertical viewing direction due to its principle and can only capture the surface with a deviation of only a few degrees. For this reason, stronger overhangs in the surface, which occur, for example, at exposed aggregates, cannot be captured by the microscope system and gaps occur at these parts in the digitized surface (see Figure 7). Consequently, these parts of the surfaces cannot be considered for evaluating the accuracy of our measurement system. 

Contrarily, our presented measurement system is able to capture slight overhangs in the surface, since the direction of the image rays is not strictly limited to the perpendicular direction. However, at these locations, the accuracy of triangulated object points is generally worse due to poorer imaging conditions. These points, however, could not be used in our conducted accuracy examinations, particularly in the performed point-to-mesh deviation computations, since these parts are incomplete in the reference meshes acquired by the microscope system—due to the above-mentioned reason. This, finally, leads to the fact that the accuracy of our measurement system is basically estimated slightly more optimistically. However, we think that this influence has a rather minor effect on the overall accuracy evaluation and is, therefore, negligible for the most part. 

Additionally, the accuracy of our proposed system was evaluated computing point-to-mesh deviations between reconstructed point clouds and corresponding reference meshes. However, it has to be considered that meshes are generally only an approximation of the actual surface, though a very high-resolution and accurate one. Unlike the true surface, which naturally has a smooth shape, meshes contain many small planar triangles that can approximate the true surface only to a certain degree. However, we assume that due to the particularly high resolution of the microscope system and, thus, correspondingly high number of triangles, the reference meshes approximate the surfaces sufficiently well and this influence, thus, becomes negligible. 

Even though all three deployed methods (sand patch method, laser triangulation and our camera-based system) provide different parameters for roughness, the empirically determined correlation coefficients of greater than 0.99 in the comparative study indicate extraordinarily high linear relationships between all three methods. Among them, the greatest correlation is between laser triangulation and the camera-based system, which is most likely due to the fact that both are based on an optical measuring principle and additionally provide in general much more accurate and reproducible results compared to the sand patch method.

## 7. Conclusions

### 7.1. Summary

In this paper, we introduced a novel camera-based measurement system for the roughness determination of concrete surfaces and provided comprehensive examinations for its measurement quality. The camera-based system mainly consists of a mechanical cross slide, which guides a monocular industrial camera over the surface to be measured and captures strongly overlapping images. The images are utilized for reconstructing 3D point clouds of the surface, which in turn are used to derive roughness parameters. 

For evaluation of the measurement accuracy, two kinds of examinations were conducted on three concrete specimens: Trueness was assessed by comparing surface measurements of the camera-based system to those of a highly accurate microscope system based on focus variation as reference. Point-to-mesh deviations between the point clouds of the camera-based system and the meshes of the microscope system revealed RMSE values of 40–50 µm.Precision was assessed by considering the scattering of multiple roughness measurements under repeat conditions. The standard deviations of the measurement series led to values less than 6 µm. Furthermore, point-to-point distances between repeatedly reconstructed point clouds of the same surfaces did not show any noticeable irregularities, apart from slightly higher deviations at the borders of the exposed aggregates, which is contributed to poorer imaging conditions at these regions.

To confirm validity, a comparative study was conducted to established roughness measurement systems using sixteen concrete specimens: The correlation coefficients between roughness measurements of the camera-based system, the sand patch method and laser triangulation led to values of higher than 0.99, indicating an extraordinarily strong linear relationship between all three methods. Among them, the strongest linear relationship was between the camera-based system and laser triangulation with a correlation coefficient of 0.9934.

### 7.2. Outlook

Currently, we have implemented only the roughness parameter *Sa*, since it belongs to the widely used ones and is fairly simple to implement on 3D point clouds. However, to enable a more direct comparison to the sand patch method and to the laser triangulation system, equivalent parameters should be implemented for our system. In addition, from some points of view, the parameter *Sa* is inappropriate to specify the respective characteristics of a surface regarding roughness. There are other parameters that can reflect relevant features of the surface topography more appropriately.

## Figures and Tables

**Figure 1 sensors-22-04211-f001:**
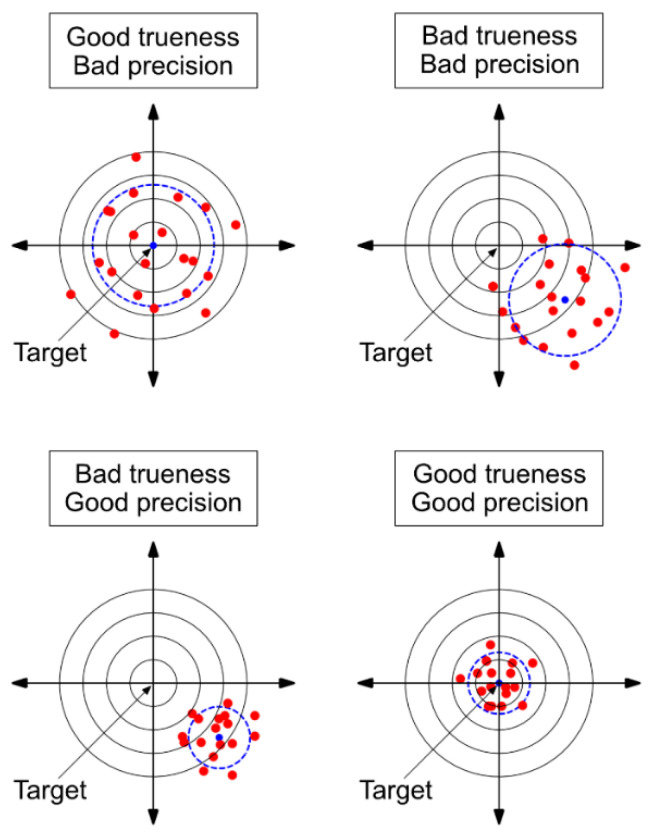
Bull’s eye for illustrating the accuracy (adapted from ISO 5725-1 [20] with permission from the German Institute for Standardisation (DIN)).

**Figure 2 sensors-22-04211-f002:**
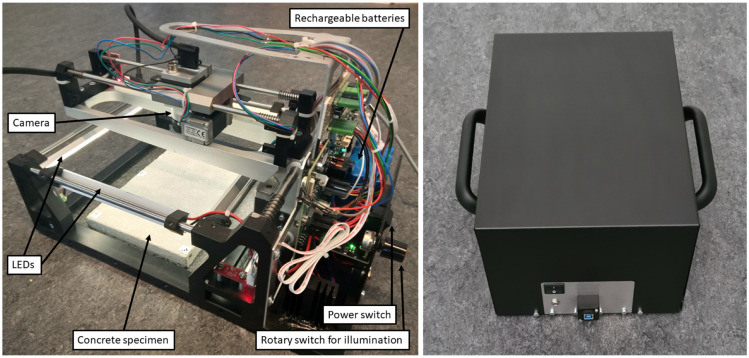
The measurement system—to the left the hardware assembly and to the right the casing.

**Figure 3 sensors-22-04211-f003:**
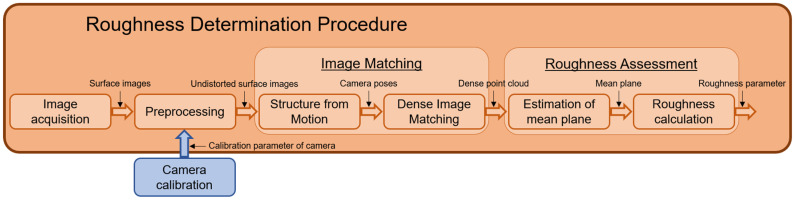
Schematic of the roughness determination procedure.

**Figure 4 sensors-22-04211-f004:**
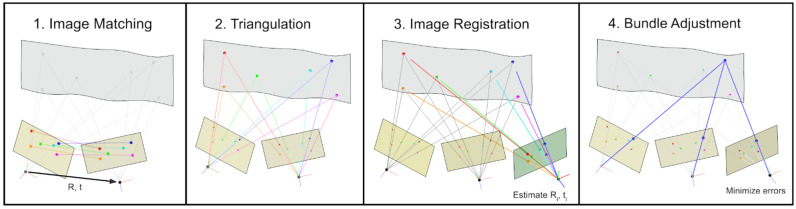
Schematic of the Structure from Motion procedure.

**Figure 5 sensors-22-04211-f005:**
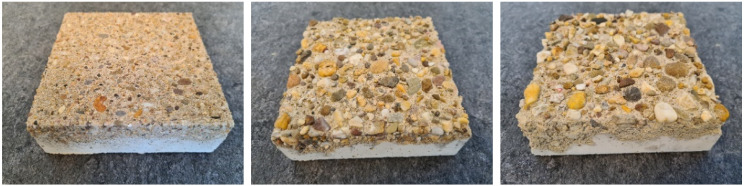
Overview of the three concrete specimens used as investigation objects for accuracy evaluation—to the left specimen 1 with 1.5 mm washout depth, in the center specimen 2 with 3.0 mm washout depth and to the right specimen 3 with 6.0 mm washout depth.

**Figure 6 sensors-22-04211-f006:**
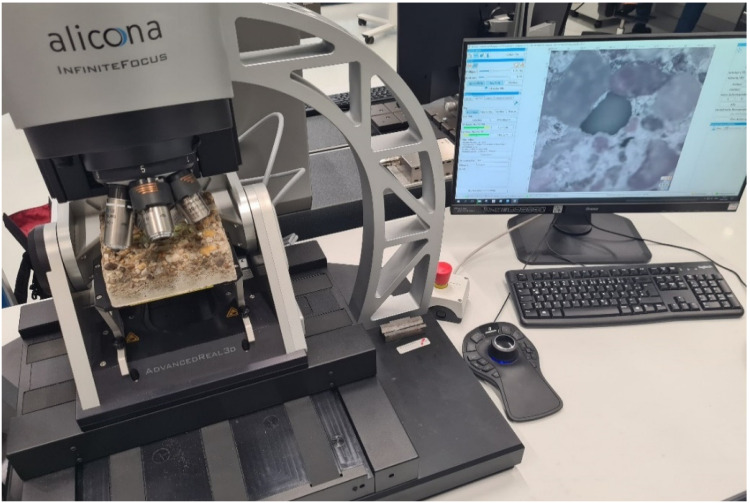
The 3D scanning microscope during the measurement of a particular concrete specimen.

**Figure 7 sensors-22-04211-f007:**
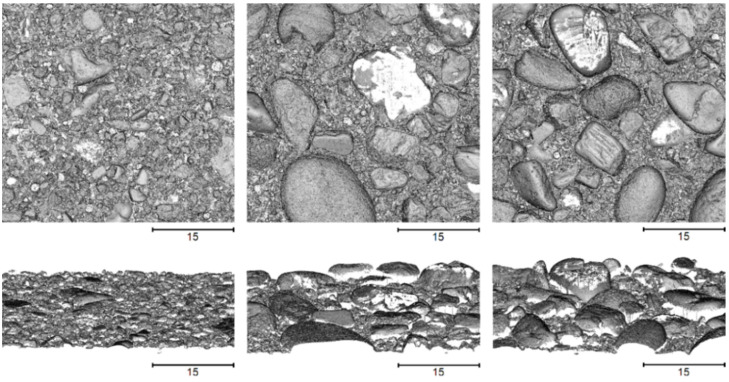
Captured surface meshes of the three specimens using the microscope system, in top and oblique view—to the left specimen 1, in the center specimen 2 and to the right specimen 3.

**Figure 8 sensors-22-04211-f008:**
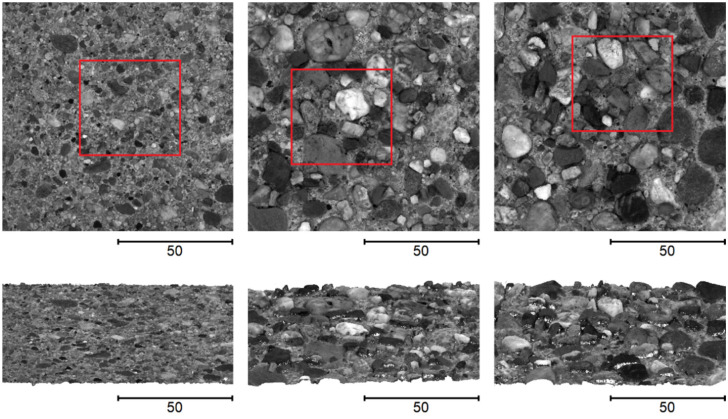
Reconstructed point clouds of the three specimens using our camera-based system, in top and oblique view—to the left specimen 1, in the center specimen 2 and to the right specimen 3.

**Figure 9 sensors-22-04211-f009:**
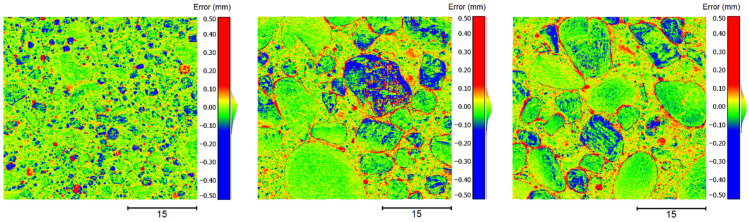
False color maps depicting the point deviations of the point clouds to the reference meshes—to the left specimen 1, in the center specimen 2 and to the right specimen 3.

**Figure 10 sensors-22-04211-f010:**
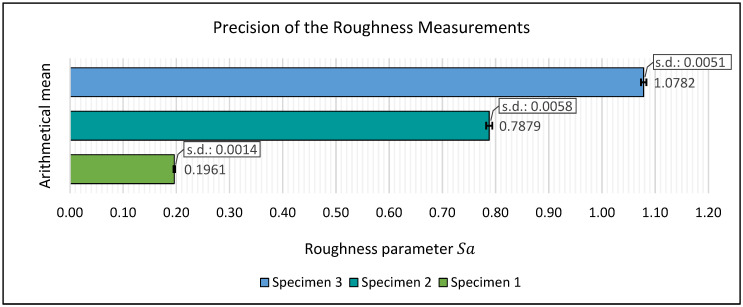
Results for the precision of the roughness measurements.

**Figure 11 sensors-22-04211-f011:**

False color maps depicting the point distances of pairs of point clouds from same specimens—to the left specimen 1, in the center specimen 2 and to the right specimen 3.

**Figure 12 sensors-22-04211-f012:**
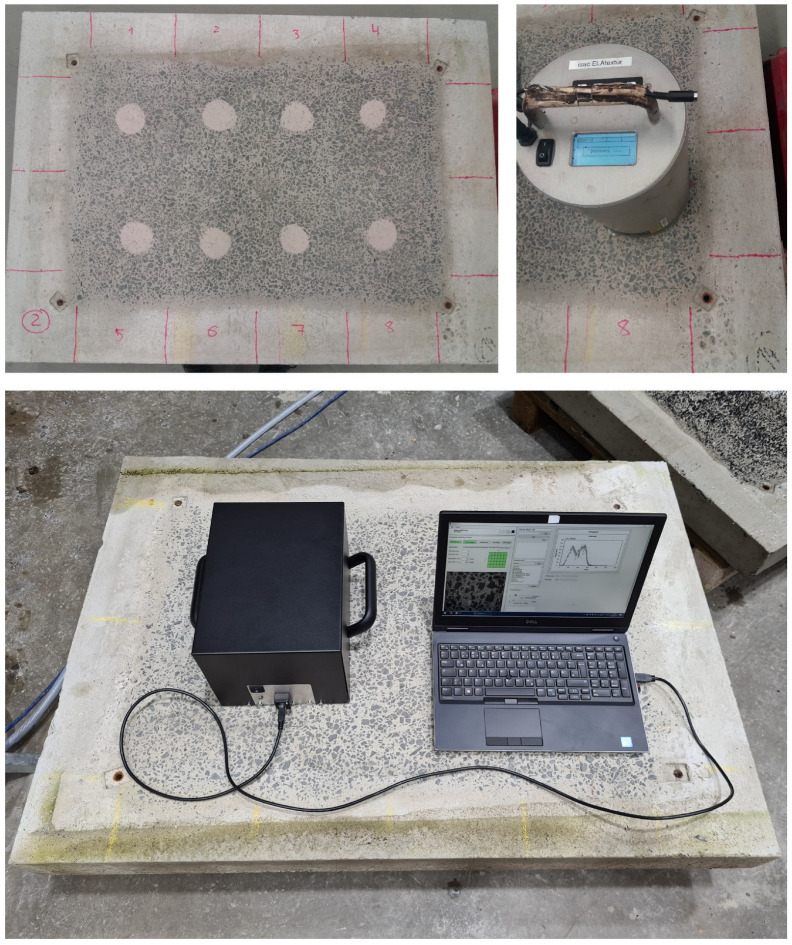
Exemplary measurement setups with all three methods—at top left the sand patch method, at top right the laser triangulation system and at the bottom the camera-based system.

**Figure 13 sensors-22-04211-f013:**
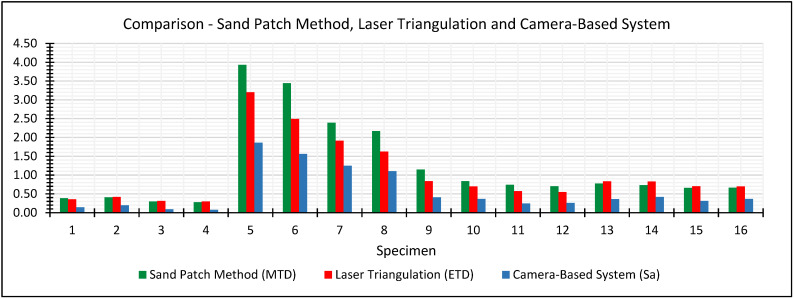
Bar chart of obtained roughness values for the 16 concrete specimens using the sand patch method, laser triangulation and our camera-based system.

**Figure 14 sensors-22-04211-f014:**
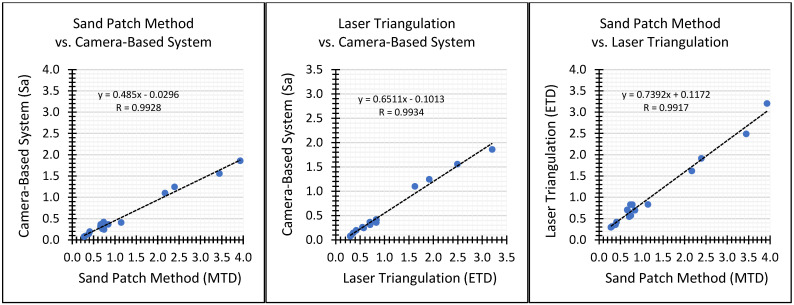
Correlations between the measurement series of the three methods deployed on the 16 concrete specimens—left sand patch method and camera-based system, center laser triangulation and camera-based system and right sand patch method and laser triangulation.

**Table 1 sensors-22-04211-t001:** Specifications of the three concrete specimens used as investigation objects for accuracy evaluation.

	Concrete Mixture			Roughening Procedure		
Specimen Number	Compressive Strength Class	Aggregate	Grain Fractured/Round	Type of Surface Retarder	Application Procedure	Wash-Out Depth
1	C20/25	Quartz	round	S3/10	negative	1.5 mm
2	C20/25	Quartz	round	S3/80	negative	3.0 mm
3	C20/25	Quartz	round	S3/300	negative	6.0 mm

**Table 2 sensors-22-04211-t002:** Specifications of the deployed 3D scanning microscope system.

Alicona InfiniteFocus G5 with 5× Objective	
Working distance	23.5 mm
Lateral measurement range (X, Y)	2.82 mm
Sampling/Measurement point distance	1.76 µm
Finest lateral topographic resolution	3.51 µm
Min. repeatability/Measurement noise	120 nm
Vertical resolution	410 nm
Vertical measurement range	22.5 mm
Vertical scanning speed	3000 µm/s
Measurement speed	≤1.7 million points/s

**Table 3 sensors-22-04211-t003:** RMSE for the point clouds using 50,000 random sample points.

Point Cloud of Specimen 1 (Fine-Grained)	Point Cloud of Specimen 2 (Medium-Grained)	Point Cloud of Specimen 3 (Coarse-Grained)
0.0416446 mm	0.0491547 mm	0.045598 mm

**Table 4 sensors-22-04211-t004:** Specifications of the 16 concrete specimens used as investigation objects for validation.

	Concrete Mixture			Roughening Procedure	
Specimen Number	Compressive Strength Class	Aggregate	Grain Fractured/Round	Surface Treatment	Pressure [bar]
1	C20/25	Basalt	fractured	sand-blasting	4
2	C20/25	Quartz	round	sand-blasting	4
3	C50/60	Basalt	fractured	sand-blasting	4
4	C50/60	Quartz	round	sand-blasting	4
5	C20/25	Basalt	fractured	water-jetting	1700
6	C20/25	Quartz	round	water-jetting	1700
7	C50/60	Basalt	fractured	water-jetting	1700
8	C50/60	Quartz	round	water-jetting	1700
9	C20/25	Basalt	fractured	concrete milling	-
10	C20/25	Quartz	round	concrete milling	-
11	C50/60	Basalt	fractured	concrete milling	-
12	C50/60	Quartz	round	concrete milling	-
13	C20/25	Basalt	fractured	stock chiseling	-
14	C20/25	Quartz	round	stock chiseling	-
15	C50/60	Basalt	fractured	stock chiseling	-
16	C50/60	Quartz	round	stock chiseling	-

**Table 5 sensors-22-04211-t005:** Results of the roughness values for the 16 concrete specimens obtained using the sand patch method, laser triangulation and our camera-based system.

Plate Number	Sand Patch Method	Laser Triangulation	Camera-Based System
	MTD [mm]	ETD [mm]	Sa [mm]
1	0.383	0.358	0.144
2	0.408	0.419	0.197
3	0.297	0.313	0.089
4	0.277	0.297	0.077
5	3.928	3.203	1.86
6	3.442	2.491	1.558
7	2.392	1.915	1.246
8	2.17	1.621	1.101
9	1.143	0.835	0.41
10	0.838	0.698	0.365
11	0.743	0.574	0.247
12	0.705	0.547	0.261
13	0.778	0.831	0.359
14	0.732	0.829	0.421
15	0.658	0.705	0.312
16	0.665	0.698	0.365

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
