# Peer review of "Quality Assessment of a Novel Camera-Based Measurement System for Roughness Determination of Concrete Surfaces—Accuracy Evaluation and Validation"

_sensors, 2022, doi:10.3390/s22114211_

Round 1

Reviewer 1 Report

This research evaluates a camera-based measuring system for evaluating the roughness of concrete surfaces, which has some promising implications for the quick detection of surface quality. However, the writing style falls short of readers' expectations. Furthermore, the major conclusion and trend analysis are unclear.

  1. The abstract and keywords are too long to read and thus should be condensed to highlight essential topics and areas of innovation.
  2. Because the key parameters of the camera-based measurement system are not shown in the paper, it is impossible to determine whether the collected images are reliable or not. For instance, the depth of field (DOF) for the camera is critical for the imaging process; does the DOF cover the washout depth of the specimen in Figure 4?
  3. The roughness calculation processes do not include a schematic.
  4. Figure 11 should be replaced with a schematic that describes each component of the system in detail.

Reviewer 2 Report

Dear author(s), please find some comments on the manuscript ‘Quality Assessment of a Novel Camera-based Measurement System for Roughness Determination of Concrete Surfaces – Accuracy Evaluation and Validation’, Manuscript ID: sensors-1725971:

  1. Abstract, even interesting and containing all of the (required) relevant information, is too long and, simultaneously, makes the reader boring. Please try to emphasize the most important studies and proposals instead of a detailed description.
  2. The “introduction’ section is appropriate, however, a more critical review would be preferred that in its current form, it is difficult to define the novelty based on the motivation.
  3. The author(s) mentioned in a sentence ‘In the following, we survey a selection of the most recent developments and most relevant research regarding surface roughness assessment and highlight their features.’, lines 120-121, that they are considered on the most recent developments but, respectively, only 5 from 25 references are from the last 5-6 years. Moreover, there is no paper from the recent journal (Sensors) but more from others, e.g. Construction and Building Materials.
  4. Dividing (roughly) the surface roughness devices, the contact methods and their limitations are widely (enough) presented in lines 128-150 but, respectively, non-contact approaches (especially their limitations) were presented too shortly. There was no review, except for gross errors, on measurement uncertainty or noise. There is some word about measurement repeating in a sentence ‘For accuracy evaluation in practice, the expected value can be approximated by the arithmetic mean of measured values under repeat conditions.’, lines 227-228. Please look for some examples:
  • https://doi.org/10.1088/2051-672X/3/3/035004
  • https://doi.org/10.1007/s41871-020-00057-4
  • https://doi.org/10.3390/ma14175096
  1. There are many shortcuts and abbreviations so, respectively, additional sections (e.g. ‘Abbreviations’) would be required.
  2. Why a mean plane was calculated if a profile (Ra) parameter was calculated and, correspondingly, considered? Looks like the author(s) are trying to analyse the surface topography with an areal performance but with a profile examination. Please look for the sentence: ‘In the case of a surface profile, this parameter basically describes the mean deviation of the profile line to a mean line.’, lines 299-300.
  3. Considering the sentence ‘In this method, for each point in the compared point cloud, the closest point in the reference point cloud is searched and the Euclidean distance is calculated.’, lines 470-471, why the Euclidean distance was calculated instead of, e.g., least square method?
  4. In Figures 8 and 10 there are no height scale units.
  5. The units should be unified in all of the subfigures in Figure 13 for both (horizontal and vertical) axes.
  6. Considering the whole ‘Discussion’ section, more critical with putting a greater impact on the limitations of the methods proposed, would be welcome. Limitations were not presented appropriately.
  7. The ‘Conclusion’ section should be more precise and, respectively, its length must be reduced. Both detailed and general conclusions should be presented each in separate and numbered gaps.

Moreover, some additional, editorial issues must be mentioned:

  1. In lines 432-433 Table 3 should be mentioned instead of ‘:’.
  2. In lines 522-523, the volume should be presented in grams?
  3. The ‘Reference’ section should be unified according to the journal template requirements.

Concluding, the manuscript is interesting, however, in its current form, is not suitable for publication in a quality journal as the Sensors is but, respectively, must be improved before any further processing. 

Round 2

Reviewer 2 Report

Dear Authors, according to the review of the revised manuscript ‘Quality Assessment of a Novel Camera-based Measurement System for Roughness Determination of Concrete Surfaces – Accuracy Evaluation and Validation’, Manuscript ID: sensors-1725971, it was found appropriately improved and, respectively, in my feelings, can be considered for publication in the quality journal as the Sensors is.

Thank you for both taking into consideration the suggested remarks and full responses, all of them were presented in a required, significant manner.

From the above, the manuscript can be further processed by the Sensors editorials.